# Value Creation through Corporate Sustainability in the Port Sector: A Structured Literature Analysis

**Michael Stein \*** and **Michele Acciaro**

Hapag-Lloyd Center for Shipping and Global Logistics (CSGL), Kühne Logistics University,
Großer Grasbrook 17, 20457 Hamburg, Germany; michele.acciaro@the-klu.org
**\*** Correspondence: michael.stein@the-klu.org; Tel.: +49-040-328707-0

**Abstract:** Corporate Sustainability (CS) in the port sector has emerged as an important driver behind strategy definition for port authorities globally. It has been argued that CS practices have the potential of delivering value for port users and, as such, grant port operators and port managing entities competitive advantages. There is, however, limited evidence behind this claim. The difficulty with collecting such evidence is that we lack measures of port value creation, and CS metrics have rarely been developed and applied in ports. This paper provides a framework for collecting empirical evidence aimed at assessing in what way CS can benefit port competitiveness. The framework is built on a systematic literature analysis of the past years. The literature analysis exceeds previous comparable contributions by its analytical detail and provides valuable new insights on sustainability in the maritime domain. The research indicates that the accurate measurement of CS initiatives in the port sector is urgent and meaningful. When appropriately measured, the value that CS can deliver to port users becomes apparent. This is, however, often created indirectly via branding, risk mitigation, etc. The paper contributes to academic knowledge as it is the first to develop a rigorous CS measurement framework usable for ports in terms of value.

**Keywords:** corporate sustainability; green ports; scale development; stakeholders; corporate social responsibility

## 1. Introduction

Ports have been working on improving their sustainability profiles in the last decades, and few ports globally can afford today to ignore the negative impacts associated with the economic activities taking place within or in the proximity of the port. As awareness for environmental and social issues increases globally, port sustainability is not a matter for developed countries only, although there are still the major ports in Europe, Asia, and North America that lead the way on environmental and social issues. Climate change has also played an important role in creating awareness of sustainability practices in the maritime sectors [1]. The increasing presence of green marketing and promotion among major ports, for example, is a sign of the perceived need of port administrators to profile the port in the eye of an increasingly critical public opinion. Although there is a great variety in the degree of commitment towards sustainability among ports [2], there is a general tendency towards making sure that everything good the port does in terms of reducing its negative impacts is publicized.

In addition to the materials made available on the Internet, port authorities and other port managing entities have developed strategic documents and sustainability reports that show how port managers intend to develop the port further and what has been achieved in the last years. Some of these documents are probably more exercises in public relations than hard commitments towards sustainability, but their publication has contributed to facilitating and informing the broader debate on

sustainable ports. While this debate is far from coming to a close, the increasing attention to enhancing sustainability in ports is certainly a positive development.

This tendency has generated a renewed interest among both academics and port specialists in the processes and drivers behind the formulation of such sustainability strategies, their effectiveness, and their impacts on port management. Although environmental and social issues among ports are not a new topic, it has been suggested elsewhere [3] that the overall deregulation that has characterized the port industry in the last decades has certainly increased the value associated with developing corporate social responsibility (CSR) among port authorities and, consequently, the need for port-specific CSR strategies.

Over the last decade, growing interest in corporate social responsibility as well as environmental impacts of and within the maritime sector increased due to pressing global concerns related to climate change and citizen mobilization on port- and shipping-related issues [4]. The media coverage of environmental issues, such as oil spills, and protest action, such as in the case of port workers' strikes, continue to maintain the visibility of port sustainability issues to the general public and have initiated questions of accountability [5]. Port managing corporations have been forced to start shifting their main objectives beyond profit maximization to include sustainable performance [6]. It is regarded as necessary to extend stakeholder involvement in maritime governance and to embrace a larger sustainable co-operation between stakeholders in the shipping industry [7,8]. In addition, buyer-driven environmental upgrading is being increasingly recognized as important, although it is not likely to result in change in management practices and operation unless it is supported by clear, predictable, and enforceable global regulations [9].

Port economic activities generate a wide range of external environmental and societal effects [10]. Port authorities are demanded to take action to minimize the negative impacts on their communities as well as society in general, and strive to maximize the value generated by port activities. In many ports around the world, port authorities are also responsible for the development and implementation of port expansion plans, and the assessment of the benefits and costs associated with such expansions is critical. Port authorities generally also act as landlords, and they exert a great influence on the definition of the terms of concession agreements and in the provision of incentives for terminal operators and port users. It is, therefore, understandable that they are often obliged to take responsibility for social and environmental effects deriving from port activities and that they should closely regard such impacts.

Ports are the locations of a variety of environmental effects, some of which derive from the nature of port business itself; others stem from the proximity of ports to urban and industrial sites, and others are the results of the specific topographies of port areas at the intersection between water and land. A distinction can be made between natural and anthropogenic pressures for ports [11]. These pressures often result in conflicts on port resource utilization, primarily land and water, which include commercial cargo loading and unloading operations, industrial activities, tourism, fisheries, and nature preservation. Given the scarcity in many regions of the world of port areas and the high costs of developing new ports, these conflicts are likely to increase in relevance over time.

Pressure to improve sustainability among production and distribution of goods has raised new challenges in all stages of the supply chain and in most industries. Nowadays, ports find themselves in the position to balance commercial and economic growth on the one side and the reduction of negative environmental and social effects on the other side. Ports, as part of a supply chain network, are required to deal with short-term views, private and public interests, and commercial and social objectives, as they are considered responsible for a wider set of environmental impacts [12,13].

Port authorities have, then, an important part to play in the moderation and resolution of such conflicts. They need to safeguard the commercial and economic interests of the port, but, given the public–private character of many port authorities around the world, are also entrusted with protecting the interest of the public and of the local port communities, on which, in the end, their agency depends. The management of stakeholders can be considered as one of the main tasks of port authorities [12].

However, little is known when it comes to measurement and quantification sustainability in port operations. It is the intention of this paper to shed light on port sustainability measurements by developing a framework to collect and benchmark empirical indicators for sustainability in ports. Existing environmental rating schemes in the shipping sector are often unclear and inconsistent in their data collection, thus creating difficulty in providing uniform measures across a very heterogenous industry [9].

This paper builds on the necessity of structuring data collection processes in port sustainability. After a brief introduction on the definition and main aspects regarding CSR in the maritime industry in Section 2, the research methodology is presented in Section 3. The methodology is split into a literature review as well as a framework analysis. Section 4 presents a conceptual framework for sustainability data collection in ports based on the literature. Section 5 concludes and discusses the limitations of the research.

## 2. Sustainability and Corporate Social Responsibility

### 2.1. Definition of Sustainability

Various contributions on the topic of sustainability in the industry vary with their specific definitions of the topic. The most widely accepted basic definition follows the so-called Brundtland Commission or World Commission on Environment and Development (WCED) of the United Nations in 1987. Sustainable development was defined as "development that meets the needs of the present without compromising the ability of future generations to meet their own needs". In other words, it was later adopted as "increasing the welfare of the present generation while simultaneously not decreasing the welfare of the next generation" [14]. Following Elkington [15], the context of sustainability was understood as a holistic concept comprising the three unique aspects of economic, environmental, and social sustainability, which is often referred to as the triple bottom line (TBL). He connected sustainability to the process of simultaneously achieving three inter-linked goals—economic prosperity, environmental protection, and social equity. This combination of three inter-linked aspects (economic, environmental, and social) was adopted by the United Nations Commission on Sustainable Development (UNCSD) in 2011, and is widely accepted in the literature of the past decade (see, for example, [1,6,16,17]). It is pointed out by Lu et al. [16] that the difference between the terms "sustainability" and "green" is significant. Although the terms are often used interchangeably, "sustainability" needs to include the consideration on economic, environmental, and societal issues, while "green" is focused only the environment. It should be stressed that when addressing any economic, environmental, or societal issue, one is rapidly confronted with their interrelations. From a maritime viewpoint, the concept of the green port was initially proposed in 2009 during the United Nations Climate Change conference, according to Wang et al. [18], and primarily focused on low-carbon emission ports. The concept of the sustainable port appeared in the literature later (e.g., [3]).

### 2.2. Perception of Sustainability

As already pointed out, port infrastructure, operations, and port-related industrial and economic activities have adverse consequences on the environment and are held responsible for negative external effects [10,19]. Ports facilitate commercial and economic growth on the one hand, but also reduce the quality of air and marine water in their vicinities on the other hand [13].

It has been highlighted that sustainability is increasingly seen as one "key driver in port development for the next decades" [1]. It is stated that ports must "plan and manage their operations and future expansion (growth) in a sustainable way in order to cope with the limited or decreased environmental space and intensified interactions with their hinterlands" [1]. On the port management side, CSR management strategies are, therefore, moving from a cost-saving orientation towards resilience and a value-adding sustainability-focused regime [1].

Environmental impacts of the shipping industry are perceived as more and more severe, including air pollutant emissions, oil and chemical water pollution, litter, sewage, and invasive species in ballast water [20]. Furthermore, the abuse of maritime policies with the use of flags of convenience to avoid national or regional regulation and tax evasion is characteristic for this industry [8]. Yliskylä-Peuralahti et al. [21] characterize two types of companies competing with each other in the shipping market: "Those companies that are responsible and focus on high-quality shipping and those that focus on providing low-cost services at the expense of safety and the environment". However, with increasing customer awareness, NGO campaigns, and emerging regulations, both national and international, aiming at enhancing the environmental impact of production and transportation of goods, the whole shipping sector (as well as other corporate actors) is driven to address the footprint linked to their transport service.

First, shipping companies and ports do realize the competitive advantages of sustainability as an instrument to enhance service quality as part of the company's differentiation strategy [22]. Unfortunately, globalization, the competitive maritime environment, and its weak regulatory frameworks led to a situation where responsible shipping companies often stand in a lower competitive position relative to companies focusing on short-term gains. This aspect is enhanced especially when non-sustainable companies diffuse CSR practices within the industry by co-operating with each other in alliances [8]. In addition, the lack of enforcement mechanisms and missing stakeholder pressure led to a relatively low number of shipping companies and ports participating in CSR practices so far.

On the regulatory side, regulations take a long time until coming into force, which often reduces the necessity for port operators to act, thus slowing down changes. Frankel [23] already included the impact of ballast water on port design in the 1980s. It is also stated that "the impacts on surface water quality are caused by generated sewage, bilge wastes, sludge, waste, oil discharges, and leakages of harmful materials both from shore and ships" [13]. After two decades of complex negotiations between IMO (International Maritime Organization) Member States, the International Convention for the Control and Management of Ships' Ballast Water and Sediments (BWM Convention) was finally adopted in 2004. Within this scope, the "Guidelines for approval of ballast water management systems (G8) have been revised in 2016 and converted into a mandatory Code for approval of ballast water management systems (BWMS Code), which was adopted by MEPC 72 (April 2018) and enters into force in October 2019" (IMO, 2019).

With slow and heterogeneous regulations on the one side and high competition on the other side, a growing number of contributions consider a pressure from the industry and non-financial stakeholders as well as customers and other institutions, such as banks, as relevant [8,9].

### 2.3. Gains from Implementing Corporate Social Responsibility

CSR is nowadays connected to a variety of advantageous factors in the maritime domain, with a growing number of indicators in the literature. On the social/ecological side, it was indicated that port-authority-driven environmental efforts raised the positive image of the local community, thus building trust in the port [24]. Without doubt, the economic aspect of CSR will have the most weight in a company's decision to change or adopt responsible measures. Studies also indicate a positive correlation between CSR efforts and economic advantages. CSR in shipping is claimed to provide an added advantage for firms by differentiating their services, avoiding port state interventions, receiving permissions to operate in environmentally sensitive areas, and improving the image for recruiting new personnel [25,26]. It is furthermore shown by Drobetz et al. [27] that "responsible firms, which contribute both economically and ethically to the society and local communities they serve, are better positioned to grow in terms of reputation and revenues". According to the Porter hypothesis [28] that was transferred to maritime sustainability by Cheon et al. [29], "stringent environmental policies and regulations can facilitate firms' efficiency and innovation, thus contributing to their ability to accomplish various sustainable development objectives". In addition to this rather general statement, more narrow aspects supporting this theory in the shipping industry were found. One example for

financial CSR gains is shown by Drobetz et al. [27], stating that an increased CSR transparency lowers information costs on the investor's side, leading to potential positive financial effects. In addition, decreased environmental incidents within a port reduce damage rates, benefitting the port's service reputation and attracting more customers [29]. Ref. [6] reveals that socially responsible activities among shipping firms will positively affect customer satisfaction, which appears to be related to public recognition of the firm. Their results imply that a shipping company facing tight competition can have a competitive edge if it satisfies its customers, since this will result in customers' long-term commitment and loyalty [6]. The case study of Wilhelm Wilhelmsen indicates how CSR rationales are already adopted by firms, including a variety of economically advantageous aspects, such as "managing risk, improving resource efficiency and access to capital, responding to or pre-empting regulations, encouraging innovation, and building future market opportunities" [30].

### 2.4. Challenges of Introducing Corporate Social Responsibility in the Maritime Sector

Despite being the most important cargo transport mode in terms of cargo numbers, the maritime transport sector is still "perceived as one of the laggards in processes of environmental upgrading" [31]. A number of reasons have been identified in the literature, indicating a certain challenge in introducing sustainability into this domain. When describing the maritime domain, one must differentiate between the shipping and the port side; however, understanding each individual challenge is necessary for a holistic understanding.

The ship owner side is affected by the challenge of highly cyclical markets with small margins [9], where the demand on transport services is derived from a variety of micro- and macro-factors in different producing industries. While many maritime studies seem to disregard this aspect by only focusing on one shipping niche, Ref. [9] found evidence suggesting a more differentiated evaluation on shipping sectors with regard to sustainability. According to them, the large shipping segments of dry bulk, tankers, and containers vary in their characteristics of relationships between cargo owners and transport service providers (shipping companies). They name "differences in type of cargo, trade distribution patterns, market concentration and ownership, contract length, and bargaining power dynamics" as reasons for their assumption. According to their research, the container market ships branded goods (container), where cargo owners start placing demands on shipping companies about their environmental performance. In tanker shipping, where oil-producing corporations represent the transport-service-requiring firms, environmental concerns about oil spills are present, as those generate high costs and damage the customer's brand due to wide media coverage.

In contrast, the dry bulk shipping market has minimal to no interest in environmental performance, as raw materials are further processed and not linked to any end customer. Consumer pressure in the bulk shipping segment is perceived as secondary given the business-to-business nature of this transport industry, with low media visibility on its environmental impact [7,9]. According to Poulsen et al. [9], "without the explicit governance traits of either strong buyer or supplier power, environmental upgrading is fundamentally absent in dry-bulk shipping". Compared to other shipping industry sectors, the cruising industry provides a different example of consumer pressure with increasing demand on CSR practices [8]. It is shown that, among these major shipping markets, buyer-driven pressure on environmental upgrading, as a result of the cargo being directly linked to the final brand customer, is key for environmental developments in the shipping sector.

A basic challenge for shipping firms remains in offering their shipping operations, being profitable and coping with competition, and increasing their environmental footprint at the same time [32]. The shipping sector has a need to remain attractive to investors and freight customers, as well as to regulators and present and future employees, were each fraction has different demands on sustainability. Furthermore, the challenge of excess shipping tonnage in the market over the past decade further drives shipping firms to lower costs [5]. On the employer side, the "size of the shipboard crew has been dramatically reduced and the profession tends to be characterized by relatively inferior working conditions and high insecurity due to short-term contracts and a high crew turnover" [8].

On the port side, it is indicated that "while ports are certainly aware of environmental initiatives, they are only realized when they are deemed economically feasible in the short term, and have no negative implications for operational efficiency," according to Veyvar et al. [17]. They base this assumption on "immense cost pressures and customers' unwillingness to pay for environmental protection in port operations." A fundamental challenge lies in a port's inability to move due to high investment barriers in setting up its infrastructure. The "presence of ports that cannot exit the market, despite low performance, also triggers greater performance variation among ports that face strong competitive pressure," according to Cheon et al. [33]. The relatively high entry barriers of environmental investments were also highlighted by Poulsen et al. [9]. Veyvar et al. [17] summarize that "while win–win situations between multiple dimensions of sustainability are possible, it is necessary to balance the different dimensions due to trade-off situations." Industries characterized by financial pressure on service costs and strong competition are particularly faced with difficulties in justifying investments without tangible effects or immediate payoff and operability [17]. It is stated that the location of a port also affects its sustainable ability, as rural ports are faced with a requirement for investments in training and education on sustainability to deal with the scarcity of skilled personnel in rural areas.

## 3. Methodology

### 3.1. Comparison with Past Literature Reviews

Table 1 shows a comparison of literature reviews of maritime and other comparable supply chain sustainability contributions. Each review follows a systematic approach, as presented in Tranfield et al. [34] or similar contributions. Although the authors of [34] state that systematic reviews shall "minimize bias through exhaustive literature searches", a closer look into the methodology of these reviews reveals potential improvement in the criteria used for literature selection. The authors of [35] limited their sampling to contributions with a minimum of 25 citations. From a qualitative viewpoint, this method is likely to exclude potentially important contributions, especially those of the last years of the timeframe, as citations grow over time. Other contributions, such as [36–39], only conducted an abstract analysis by screening for relevant (and the most important) subjectively chosen topics.

The authors argue that both title and abstract consist of a brief description of a research, but are not enough to provide a clear and full comprehension of a contribution. Clustering contributions through a literature analysis is an important step in science, requiring going "beyond mere descriptions of the paper" [40]. The gaps revealed among the past contributed literature reviews in the area of sustainability, however, indicate that the analyses were not comprehensive or that the basics tenets of systematic literature reviews, as proposed by [34,40], were not adhered to.

**Table 1.** Comparisons among various literature review contributions.

| Publisher | Year | Scope | Timeframe | Sample | Ratio |
|---|---|---|---|---|---|
| Maritime sustainability literature reviews | | | | | |
| [35] Sislian et al. | 2016 | 198 | 1987–2013 | 49 | 0.247 |
| [39] Lim et al. | 2019 | 704 | 1990–2017 | 21 | 0.030 |
| [41] Davarzani et al. | 2016 | 2180 | 1975–2014 | 338 | 0.155 |
| [42] Bjerkan and Seter | 2019 | 148 | 2010–2018 | 70 | 0.473 |
| [43] Hakam and Solvang | 2013 | 334 | 1985–2012 | N.A. | N.A. |
| This study | 2020 | 104 | 2016–2020 | 72 | 0.692 |
| Other supply-chain sustainability literature reviews | | | | | |
| [36] Tachizawa and Wong | 2014 | 681 | 1976–2014 | 39 | 0.057 |
| [37] Centobelli et al. | 2017 | 415 | 1960–2014 | 46 | 0.111 |
| [38] Evangelista et al. | 2018 | 582 | 2000–2016 | 88 | 0.151 |
| [44] Rajeev et al. | 2017 | 1068 | 2000–2015 | 59 | 0.055 |
| [45] Aguinis and Glavas | 2012 | 588 | 1970–2011 | 181 | 0.308 |

Another crucial factor in past literature reviews is the ratio of the actual sample compared to the scope of articles reviewed. Past contributions mainly considered a ratio of only 5–16% (see [36–41]), while few contributions considered more than 20% [35,45], and only Ref. [42] considered almost half of their scope with 47%. The authors argue that disregarding the majority of contributions in a literature analysis is most likely to create bias in the results, as many disregarded contributions might have contained valuable information about the research topic. This argument is further underlined by the intangible nature of sustainability itself, creating difficulties in weighing, measuring, and comparing using statistical techniques. This aspect makes it even more crucial to maintain a qualitative view on the topic and to gather bits of information from various contributions in order to create knowledge as a whole.

### 3.2. Literature Analysis Description

The literature analysis followed the methodology of a systematic literature review based on [34,40]. The analysis aimed to identify articles published in peer-reviewed and open-access journals in the English language from no earlier than January 2016 to provide contemporary insights into the broad aspect of maritime sustainability. The authors included open-access journals, such as "Sustainability", because relevant insights into a research topic are not exclusively reserved for peer-reviewed journals, especially when research is innovative, thus lacking a certain basis of knowledge. There were several reasons for considering the short timeframe of only 4.5 years, ranging from January 2016 until June 2020.

Firstly, the analysis follows up on the work of Acciaro (2015) [3], who already provided a detailed discussion on CSR value creation in the port sector based on an extensive literature analysis. Secondly, the concept of CSR-based literature analysis in the maritime domain is not new, and has been evaluated by various contributions (see [35,36,39,41,43–45]). However, a detailed look at these contributions reveals a gap in recent contributions after 2015, as displayed in Table 2 below. Thirdly, the authors aimed to reflect the contemporary aspect of maritime sustainability. As regulations and orientations (both political and economic) change due to geopolitical events, so changes the focus on sustainability, requiring evidence based on recent contributions of the most recent years. This aspect is not chosen randomly, but follows the approach of [42] that already addressed this aspect.

**Table 2.** Comparisons among various literature review contributions.

| Year of Publication | Number of Contributions |
| --- | --- |
| 2016 | 20 |
| 2017 | 17 |
| 2018 | 22 |
| 2019 | 10 |
| 2020 (till June) | 3 |
| Sum | 72 |

The literature review was conducted using online databases by applying the following keyword structure, as shown in Table 3. All contributions were read completely before being regarded/disregarded for the final review. The contributions were checked for cross-references to studies made in the area of interest. Those cross-references were taken into account to enhance the reliability of this study's literature review. In total, 104 contributions were identified, of which 32 were excluded for not meeting the requirements. Among the excluded contributions, 20 did not contribute to the research topic in terms of content (12 "port pricing", 6 "CSR in Shipping", 1 "port sustainable pricing", 1 port incentives), 4 contributions were university-owned publications outside any journal, and 8 contributions were book chapters or conference proceedings. Only Bjerkan and Seter [42] provided a comparable level of detail by choosing their sample and including cross-references. In total, the literature research revealed 72 out of 104 contributions, resulting in a ratio of 69.2%, or a sample reduction of only 30.8%.

**Table 3.** Keyword search matrix.

| | | | | |
|---|---|---|---|---|
| CSR | | | | |
| OR | | Port | AND NOT | Airport |
| Corporate Social Responsibility | AND | OR | | |
| OR | | Shipping | | |
| green | AND | OR | | |
| OR | | Harbor | | |
| sustainable | AND | OR | | |
| OR | | Maritime | | |
| sustainability | | | | |
| OR | | | | |
| environmental | | | | |
| OR | | | | |
| green | | | | |
| OR | | Port pricing | AND NOT | Airport |
| sustainable | AND | OR | | |
| OR | | Port incentives | AND NOT | Airport |
| environmental | | | | |

The 72 considered contributions were distributed among the years 2016–2020 and are shown in Table 2. The research was conducted until June 2020, so that the annual number of related contributions could be estimated to reach a comparable number. The sample contributions are distributed among 23 journals, with the top five journals accounting for 54.1% of the sample, and are distributed among Transportation Research Part D (16), Maritime Policy & Management (10), Sustainability (6), The Asian Journal of Shipping and Logistics (5), and Marine Policy (4), as shown in Table 4.

**Table 4.** Comparisons among various literature review contributions.

| Journal | Contributions |
|---|---|
| Transportation Research Part D | 16 |
| Maritime Policy & Management | 10 |
| Sustainability | 7 |
| The Asian Journal of Shipping and Logistics | 5 |
| Marine Policy | 4 |
| Journal of Cleaner Production | 3 |
| Energy Policy | 3 |
| Maritime Economics & Logistics | 2 |
| Journal of Business Ethics | 2 |
| International Journal on Shipping and Transport Logistics | 2 |
| Environmental Science and Policy | 2 |
| WMU Journal on Maritime Affairs | 2 |
| Transport Policy | 2 |
| Geoforum | 2 |
| Research in transportation business & management | 2 |
| Marine Pollution Bulletin | 1 |
| Sustainable Development | 1 |
| Journal of Marine Science and Engineering | 1 |
| Safety Science | 1 |
| Ocean and Coastal Management | 1 |
| Transportation Research Record | 1 |
| Journal of Transport Geography | 1 |
| International Journal of Logistics Management | 1 |
| Total | 72 |

Out of the 72 contributions of the sample, 19 were of a theoretical nature, 42 of a practical nature, and 11 used a mixed-method approach. Out of the non-theoretical studies, 15 contributions were

conducted in Asia, 18 in Europe, 11 globally, 8 in North America, and 1 in Africa. The distribution of the contributions is displayed in Figure 1.

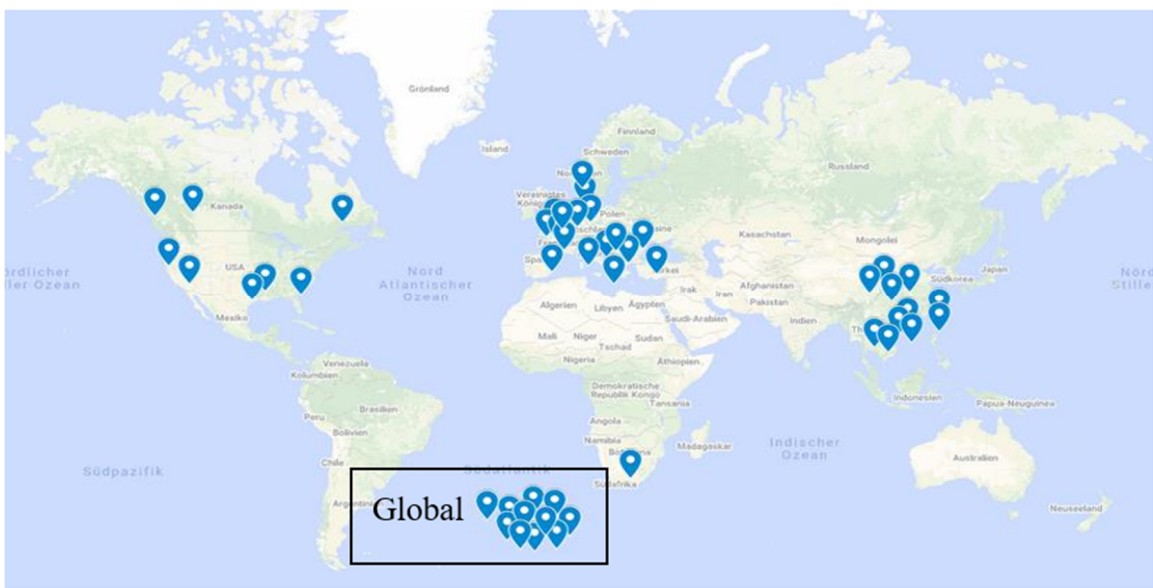

**Figure 1.** Global distribution of the sample's research. Source: Authors (2020) using Google Maps.

On the basis of the literature review, a conceptual framework was created (see Table 5). A conceptual framework is defined as an either visual or written product explaining the key factors, concepts, or variables to be studied and the presumed relationships among them [46]. Studies furthermore pointed out the importance of structured frameworks as a basic contribution for future research (see [46,47]). This paper focuses on providing a structured basis to empirically understand and measure sustainable action in the maritime domain. The combination of a conceptual framework analysis based on a structured in-depth literature analysis of contemporary contributions was, therefore, chosen to be a relevant and well-suited research tool.

**Table 5.** Framework clustering.

| Cluster | Usable Framework | Contributions |
|---|---|---|
| | Corporate social responsibility (CSR)-affecting theories | [33] |
| Underlying theories | CSR customer satisfaction theories | [22] |
| | CSR theories | [29,38] |
| | Policy initiatives and practices | [9,19,48] |
| | Port examples on environmental strategies | [17] |
| | Power/fuel topics | [42] |
| | Top 10 environmental priorities in EU ports mentioned | [49] |
| CSR policy and decisions | Policy initiatives and practices | [50] |
| | Owner alliances | [8] |
| | Ship rating schemes | [8] |
| | Green hinterland strategy matrix | [51] |
| | Instruments available to port authorities | [52] |
| | Port competition | [53] |
| | Management's perception/concerns about CSR | [5,6] |
| Affecting factors | Implementation complexity | [54] |
| | Stakeholders | [55] |

<div align="center">**Table 5.** *Cont.*</div>

| Cluster | Usable Framework | Contributions |
|---|---|---|
| Measurements | Sustainability performance measurements | [1,16,56–59] |
| | Environmental Performance Indicators | [60] |
| | | [61] |
| | | [62] |
| | Particular Matter 10 comparison of ship/shore energy sources | [63] |
| | Generic energy mapping and consumption | [64] |
| | Social sustainability indicators | [65] |

Among the 72 evaluated contributions from 2016–2020, several contributions provided individual frameworks with regard to various topics of sustainability in the maritime sector. In sum, 32 frameworks were identified and clustered into the four topics of CSR measurements (12), CSR policy and decisions (11), CSR-affecting factors (5), and underlying theories regarding CSR in the maritime sector (4). Four contributions provided a qualitative evaluation of the basic theories that affect or cause CSR actions in the maritime domain. These studies range from general CSR-affecting theory [33] over customer satisfaction theory [22] to general CSR theories [29,55] CSR policy initiatives and practices were evaluated in eleven studies from 2016–2019. Both theories and policy practices reflect a rather qualitative view of CSR in shipping, but only provide a limited capability of actually building empirical evidence regarding this topic. For the sake of this paper's research, focus will be shifted towards extracting information on how to measure CSR operations. In sum, 18 contributions provided frameworks that indicated how to measure CSR or CSR-related values in the maritime domain. Factors affecting CSR were differentiated in port competition factors [53].

The managements' perceptions of CSR [5,6], CSR implementation complexity [54], and stakeholders [55], as well as measurements regarding CSR-related factors, were evaluated among 12 studies. These studies range from sustainability performance measurements (6) over environmental performance indicators (3) to other indicators (3). A detailed survey of the identified frameworks is provided in Table 6.

<div align="center">**Table 6.** Economic factors.</div>

| Cluster | Aspect | Measurement | Source |
|---|---|---|---|
| Income and profitability | Amount of cargo handled | annual cargo volume | [56] |
| | Productivity/throughput/growth | cargo volume per vessel | [56,57,61] |
| | Corporate and property taxes | tax income | [56] |
| | Input cost | costs | [56] |
| | Investment and market share | investment amount, market share | [6,61] |
| | Management efficiency | | [56] |
| Service quality | Hinterland connection | meters of transport ways, amounts of connections | [61,66] |
| | Quality of handling | numbers of accidents, environmental impact per handling | [6,53,61] |
| | Port operations | qualitative questionnaires | [53,56] |
| | Port charges | costs | [56] |
| | Input cost | costs | [56] |
| Macro-value | GDP generation | GDP income | [56,57,66] |
| | Tax generation | tax income | [56] |
| | Trade facilitation | trade amounts | [56] |
| | Cruise tourism | passenger numbers | [61] |
| | Traffic | transhipments, cargo handling | [53,61] |

Given the limitations of this paper's research, the framework analysis focused on contributions regarding affecting factors and measurements of studies that already contributed. The TBL approach was applied to differentiate measurements into categories of economic (Table 6), social (Table 7), and environmental (Table 8) factors.

**Table 7.** Social factors.

| Cluster | Aspect | Measurement | Source |
|---|---|---|---|
| Community impact | Employment | number of jobs created | [6,56,61,66] |
| | Safety | number of safety incidents | [56,62] |
| | Security | number of security incidents | [62] |
| | Resilience | recovery time | [57,61] |
| | Heritage and cultural impact | existing Yes/No | [56,61] |
| Employment quality | CSR communication/education | quality of training | [1,5,16,58,60] |
| | CSR decision involvement | existing Yes/No | [16] |
| | Corporate culture | existing Yes/No | [16,58,67] |
| Legal and political benefits | CSR policy | existing Yes/No | [1,5,6,16,58,60] |
| | CSR information publication | number of reports | [5,16,58,68] |
| | CSR efforts beyond compliance | existing Yes/No | [1,58] |
| | Establishment of evaluation indicators | existing Yes/No | [1,58] |
| | Green port development plan | plan existing | [1,5,54,60] |

**Table 8.** Environmental factors.

| Cluster | Aspect | Measurement | Source |
|---|---|---|---|
| Water pollution management | Fuel spill contingency plan | existing Yes/No | [58] |
| | Ballast water pollutant control | existing Yes/No | [6,58,61] |
| | Cargo spill control prevention | existing Yes/No | [58] |
| | Sewage/wastewater treatment | existing Yes/No | [56–58,60,66] |
| Eco-efficiency | Hazard waste management | existing Yes/No | [56,58,62,69,70] |
| | Solid waste dumping management | existing Yes/No | [56–58,60] |
| | Energy consumption | in KW/h | [16,58,60–62,66,70–72] |
| | Water consumption | in liters | [60,62,69,70] |
| | Waste generation | in tons | [60,66,70,71] |
| | Green materials/designs for construction | existing Yes/No | [1,16,58] |
| | Heat generation | | [58] |
| | Energy quality | renewable source Yes/No | [16,54,58,63,64,68,72] |
| Air pollution management | Speed/combustion reduction | existing Yes/No | [1,58,60] |
| | Regulations on the emissions of toxic gas | existing Yes/No | [16,54,57,58,62,69–71] |
| | Cold ironing | existing Yes/No | [1,58,72] |
| | Encouraging the use of low-sulphur fuel | existing Yes/No | [1,54,58,72] |
| | Encouraging public transport mode development | existing Yes/No | [58] |
| | Light emissions | sustainable source Yes/No | [58,60] |
| | Dust control | existing Yes/No | [16,58] |
| | Emission reduction due to berth allocation | in tons | [73–77] |

**Table 8.** *Cont.*

| Cluster | Aspect | Measurement | Source |
|---|---|---|---|
| Noise control | Noise reduction | in decibels | [58] |
| | Regulations on noise control | existing Yes/No | [57,58,66,70] |
| | Avoiding disturbance to the community during infrastructure construction and expansion | | [58] |
| Marine ecological protection and biology system preservation | Wetland and marine habitat preservation | existing Yes/No | [56,58,61] |
| | Reducing infrastructure disturbance to marine biology density | | [56,58,61,70] |
| | Port entrance sediment and coastal erosion control | existing Yes/No | [58,61] |
| | Soil and sediment quality | | [62,69,70] |
| | Biotope creation | existing Yes/No | [61,66] |
| | Tree planting in port area | existing Yes/No | [16,58] |
| | Dredging sediment disposal | existing Yes/No | [58,61,66] |
| | Ballast water pollutant control | existing Yes/No | [58,61] |

Economic factors can be differentiated into clusters of income and profitability, service quality, and macro-values. While the clusters of income and profitability and service quality reflect internal economic CSR factors, the aspect of macro-values draws a broader picture. It is once more shown that ports, although being a major source of pollution and ecological disturbance, do create a benefit for their surrounding society in terms of welfare, job generation, and tourism. A holistic CSR discussion should always keep both advantages and disadvantages in mind.

## 4. Framework Analysis

Although social and environmental factors are widely covered in the evaluated literature, the individual measurements often refer to a simple existing/non-existing question in the beginning. Further research should focus on the cost structure of a port operation and its connection to port operating costs.

On the basis of the literature, the conceptual framework presented in Figure 2 is proposed. External factors include regulation, macroeconomic conditions, port governance, and societal perceptions. They influence port business activities as well as constraint port strategy. Port business generates economic, social, and environmental impacts. How port business activities generate these impacts is the result of CSR activities. The impacts also shape the CSR, which is seen as an integral part of corporate strategy. Assuming that the main objective of a port business strategy is value creation, future research should find reliable metrics of value and impacts and empirically assess how CSR actions impact value creation in the port sector.

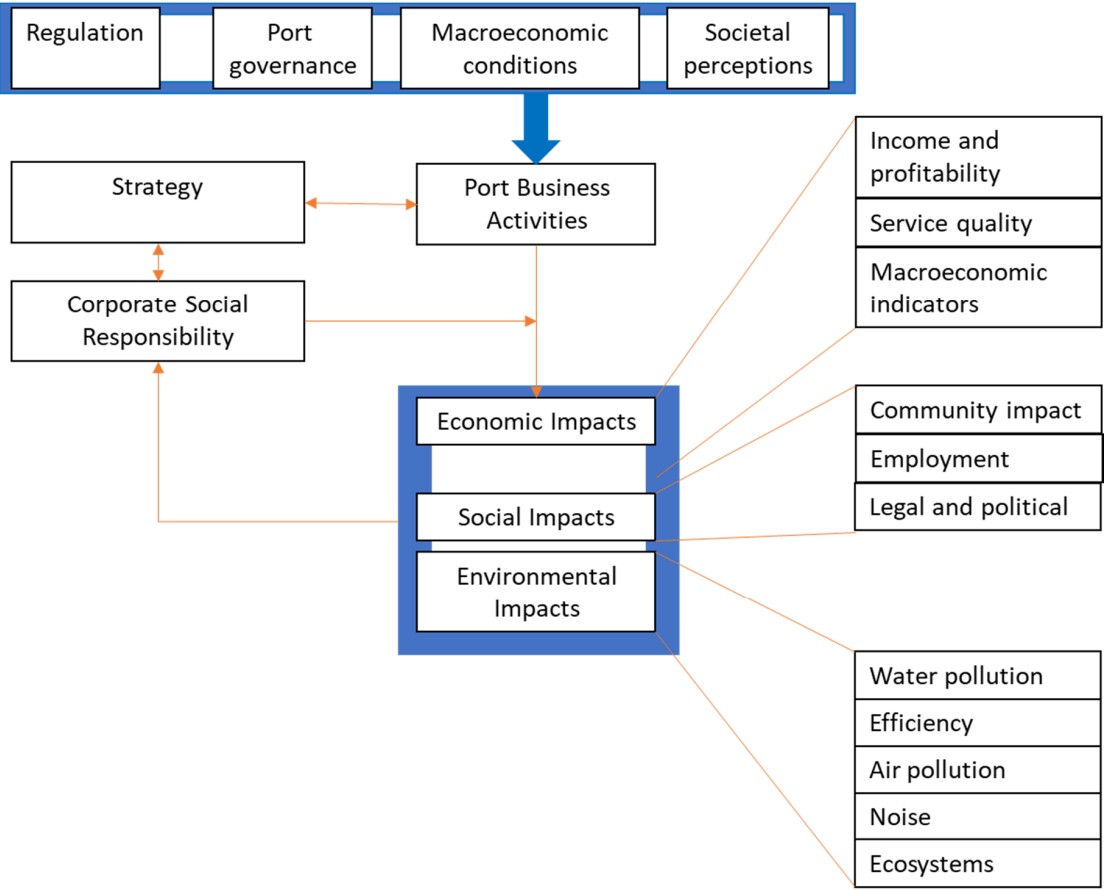

**Figure 2.** Proposed conceptual framework.

## 5. Conclusions

Building empirical evidence on an intangible asset such as sustainability is a challenging yet important task for maritime academics and for port managers. Sustainability will remain a top priority for the maritime industry, and increasing pressure will be placed on global value chains to reduce their social and environmental impacts. It is therefore important to develop structured approaches to measure the benefits of sustainability and to increase sustainability visibility in the maritime chains. This paper conducted a detailed analysis on maritime sustainability literature. A total of 104 contributions were analyzed in detail, and 16 existing frameworks developed in previous sustainability studies were connected. The framework links applicable measurements of sustainable action to port operations based on academic contributions of the past decade.

In comparison to prior literature analyses on CSR in the maritime industry as well as other transport logistics areas, the analysis conducted here exceeds previous studies through its in-depth analysis and sample size. It is the first study that sheds light on the very contemporary aspect of CSR literature after 2016, and provides new and valuable insights for academia, stakeholders, and policy- and decision-makers. The proposed conceptual framework uses the triple bottom line approach of known CSR discussions.

This paper contributes to academic knowledge, as it is the first to develop a Corporate Sustainability (CS) measurement framework that is usable for ports in terms of value creation. The paper is beneficial to society and business, since it offers a framework that can be applied in practice to measure the effectiveness of CS initiatives in terms of value for ports.

Based on the limitations of this paper as a literature analysis, future research should:

- Test and refine the proposed framework;

- Enhance the framework on the basis of similar studies in other domains;
- Determine adequate metrics to measure value and impacts;
- Develop an economic model to evaluate the relationship between port business activities and CSR;
- Test the framework by quantifying the value of CSR activities to specific port cases and then across ports.

Since CSR in the maritime domain remains an ongoing discussion in the academic context, the need for contributions that close the gap between theory and practice is of benefit for future environmental awareness within the maritime transport sector. This contribution provides a solid basis for future research on value creation and value measurement on CSR operations in ports on a contemporary basis of the most recent years.

**Author Contributions:** M.S.: Design of the study, conducting the literature analysis, interpretation of the data, conception of the paper, implementation of reviewer suggestions; M.A.: quality control, assistance in writing the paper, framework creation. All authors have read and agreed to the published version of the manuscript.

**Funding:** This research has been supported by the Social Sciences and Humanities Research Council of Canada (SSHRC) project (N∘ 895-2017-1003): "Green Shipping: Governance and Innovation for a Sustainable Maritime Supply Chain". The APC are paid by the Hapag-Lloyd Center for Shipping and Global Logistics (CSGL).

**Conflicts of Interest:** The authors declare no conflict of interest.

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
