# Peer review of "Value Creation through Corporate Sustainability in the Port Sector: A Structured Literature Analysis"

_sustainability, doi:10.3390/su12145504_

Round 1
Reviewer 1 Report
Dear Authors,
thank you a lot for the interesting paper. The subject is very up-to-date. Every enterprise from every type of industry should take into account problem of CSR and CS. It is a concept according to which enterprises at the stage of building a strategy take into account social interests and environmental protection, as well as relations with various groups of stakeholders. Being responsible not only means that enterprises meet all formal and legal requirements, but also increased investment in human resources, in environmental protection and relations with stakeholders who can have a real impact on the efficiency of our business operations and innovation.
Topic of CSR is not new, so "novelty" is average. You presented very specific and very narrow area of its use. That is why I gave average rating in "interest to readers" because it can be used and read mainly by people connected to maritime sector. "Significance of Content" and "Scientific Soundness" I assessed very high. I know how much effort you had to put into the search for literature sources and analyze them.
I found some small mistakes and I have some comments for you. This is why I gave also average rating in case of " Quality of Presentation". But your mistakes are not connected to merits of the paper but to its presentation. I hope it will help you to improve your paper.
- Line 21: Check expression "this it is often". Is it correct? I think you should remove something.
- Line 34: after [1] there is lack of dot. The same line 130 at the end.
- According to template when you quote more than 1 literature source you should write [2,3], or [4–6]. You put for example [7]; [8]. - so correct in all paper.
- When you use expression " according to", " shown by", "reveal", "found" etc... it would be better to add names of authors so reader does not have to check such info references. But also it would be nice for authors to see their names inside the paper. Especially that you use their work. There are many places like this in your paper. In other parts of the paper you put names. Better to correct it otherwise you paper seems like it was made of some parts of other papers.
- Line 138: Sentence " They state that ports.....". I don't like this sentence. they you mean who? maybe better "It is stated"?
- Lines 165-168: Check the long sentence there. You have reference at the beginning and the end of the sentence.
- Lines 298 and 305: there is error displayed. Check it.
- Line 315: in English you should use "dot" not "coma" to separate the integer from the decimal fraction.
- Since in the text you put first Table 4, later Figure 1, maybe better to insert them into paper in the same order? I think also that between Figure and 1 there is lack of space in text (line 329).
- Figure 1: reference? did you create map alone? if not, to avoid problem for you and MDPI, add source of map.
- Table 5: Last column you wrote authors- so in this table it would be better to insert authors names.
- In Table 7 you repeat Y/N . You should introduce it to readers.
- Figure 2: Is it possible to change color for black or a bit darker. I am old and for me its quality with gray color does not help.
- Check references: 23 (name of author with capital letter). Many references have "Pp", others do not have it even if you put pages. Check template of the paper and unify it.
Author Response
Dear reviewer, thank you very much for your detailed and helpful review which increased the quality of the paper by providing very valuable insights.
Your kind words at the beginning as well as your understanding for the amount of work being put into the contribution were very encouraging. The novelty of the contribution was not meant to be in the area of CSR, which clearly is not a new topic, but rather to provide new insights that CSR brings to port operations. Your comment, however, is absolutely correct and was addressed in the revision process.
Each of the comments 1-14 were addressed in the revision process. These added a great value to the paper’s quality for which we are very grateful. Please find changes made according to all reviewer’s comments in green colour in the major revised version.
Reviewer 2 Report
This study can be found to be meaningful in that it suggests a direction for sustainability in the port sector based on a comparison of past prior studies on an interesting topic. However, there is a question about what differentiates it from previous studies. Therefore, the following questions are presented in terms of differentiation from previous studies. It is a research topic that has been continuously conducted since the past, and it is necessary to clearly state the importance of why it is important to review past research. If it is simply to collect the empirical content of the port's sustainability, I wonder what differentiates it from prior research. It is not clear what the difference is from the previous studies, so I think it is necessary to fully explain and supplement it.
Author Response
Dear reviewer, thank you for your comment on the paper. The novelty of the contribution lies within the contemporary nature of the literature review after 2015 as well as the depth of the review conducted. Only two literature reviews of application of CSR in the maritime industry have been conducted after 2015 in an area that is growing very fast.
Our study is by far the most detailed when it comes to in-depth literature analysis as it not only consists of title and abstract analysis but of a 100% full paper analysis of 104 studies over the past 5 years. Further the triple bottom line clustering approach of the literature framework has not been used in previous contributions.
We took your comment seriously into consideration and changed the conclusions of the paper in order to address your concern. Please find changes made according to all reviewer’s comments in green colour in the major revised version. Thank you for your kind feedback that enhanced the paper’s quality. We thank you for your review.
Reviewer 3 Report
This is a well-written paper including both port sustainability review and a structural framework for future studies. The contributions are well documented. I have enjoyed reading it. Following comments can be addressed.
1) I would like to bring authors' attention to following relevant studies on reviewing port energy efficiency (first paper can go in Table 1,3 and 4) and sustainability reporting in inland ports;
A review of energy efficiency in ports: Operational strategies, technologies and energy management systems. Renewable and Sustainable Energy Reviews, 112, pp.170-182, 2019.
Sustainability Reporting for Inland Port Managing Bodies: A Stakeholder-Based View on Materiality. Sustainability, 12(5), p.1726, 2020.
2) Title can be written in small letters.
3) You can emphasize the contributions of reducing ship emissions at and close proximity to ports. Following optimisation studies look into reducing ship emissions considering port operations. You might note them in your paper:
The multi-port berth allocation problem with speed optimization and emission considerations. Transportation Research Part D: Transport and Environment, 54, pp.142-159, 2017.
Berth allocation considering fuel consumption and vessel emissions. Transportation Research Part E: Logistics and Transportation Review, 47(6), pp.1021-1037, 2011.
4) Figure 2's in-box texts are hard to read as they are shaded. I would recommend you to write them in black.
5) Language: I suggest;
- To delete , at the end of line 179.
- Line 298, recent contributions rather than modern
- Line 299,305 in-text reference errors can be corrected
- Paragraph 362-365 can relocate above in the text
Author Response
Dear reviewer, thank you for your kind and encouraging words in the beginning as well as bringing attention to four new contributions. Your review added great value to the paper. The textbox was changed and the language suggestions were each implemented into the paper. The titles were changed to a uniformly style of small letters. We included your comments in the major revised paper. Please find changes made according to all reviewer’s comments in green colour in the major revised version. We added the suggested paper as [71];[72];[73];[74] and are very grateful for your comments. Your feedback enhanced our framework by the aspect of berth allocation, an aspect that was not considered before (see Table 8).
You provided a very valuable feedback and helped us advancing our framework for what we are grateful.
Round 2
Reviewer 3 Report
Accept as is.